# The Survival Outcomes, Prognostic Factors and Adverse Events following Systemic Chemotherapy Treatment in Bone Sarcomas: A Retrospective Observational Study from the Experience of the Cancer Referral Center in Northern Thailand

**DOI:** 10.3390/cancers15071979

**Published:** 2023-03-26

**Authors:** Wachiranun Sirikul, Nida Buawangpong, Dumnoensun Pruksakorn, Chaiyut Charoentum, Pimpisa Teeyakasem, Nut Koonrungsesomboon

**Affiliations:** 1Department of Community Medicine, Faculty of Medicine, Chiang Mai University, Chiang Mai 50200, Thailand; 2Department of Family Medicine, Faculty of Medicine, Chiang Mai University, Chiang Mai 50200, Thailand; 3Department of Orthopedic, Faculty of Medicine, Chiang Mai University, Chiang Mai 50200, Thailand; 4Center of Multidisciplinary Technology for Advanced Medicine (CMUTEAM), Faculty of Medicine, Chiang Mai University, Chiang Mai 50200, Thailand; 5Division of Oncology, Department of Internal Medicine, Faculty of Medicine, Chiang Mai University, Chiang Mai 50200, Thailand; 6Department of Pharmacology, Faculty of Medicine, Chiang Mai University, Chiang Mai 50200, Thailand; 7Clinical Research Center for Food and Herbal Product Trials and Development (CR-FAH), Faculty of Medicine, Chiang Mai University, Chiang Mai 50200, Thailand

**Keywords:** sarcoma, osteosarcoma, Ewing’s sarcoma, chemotherapy, survival rate, prognosis, adverse effects

## Abstract

**Simple Summary:**

Overall survival for osteosarcoma and Ewing’s sarcoma has been improved since the introduction of multi-agent chemotherapy. However, there has been variability in the chemotherapy regimens for both bone sarcomas. The present study describes chemotherapy regimens being prescribed in a tertiary cancer center in Thailand and identifies prognostic factors for survival outcomes. Chemotherapy regimens for osteosarcoma vary, with the two-drug regimens (i.e., Cisplatin plus Doxorubicin and Ifosfamide plus Doxorubicin) being commonly used. The regimen for Ewing’s sarcoma was rather well established, with Cyclophosphamide plus Vincristine plus Doxorubicin alternating with Ifosfamide plus Etoposide being the most commonly used one. Response to neoadjuvant therapy and female gender may be predictive of favorable outcomes in resectable osteosarcoma.

**Abstract:**

This study aimed to assess survival outcomes, prognostic factors, and adverse events following chemotherapy treatment for osteosarcoma and Ewing’s sarcoma. This retrospective observational study was conducted to collect the data of the patients with osteosarcoma or Ewing’s sarcoma who received chemotherapy treatment between 2008 and 2019. The flexible parametric survival model was performed to explore the adjusted survival probability and the prognostic factors. A total of 102 patients (79 with osteosarcoma and 23 with Ewing’s sarcoma) were included. The estimated 5-year disease-free survival (DFS) and 5-year overall survival (OS) probabilities in patients with resectable disease were 60.9% and 63.3% for osteosarcoma, and 54.4% and 88.3% for Ewing’s sarcoma, respectively, whereas the 5-year DFS and 5-year OS for those with unresectable/metastatic disease remained below 25%. Two prognostic factors for osteosarcoma included a response to neoadjuvant chemotherapy and female gender. Ewing’s sarcoma patients aged 25 years and older were significantly associated with poorer survival outcomes. Of 181 chemotherapy treatment cycles, common self-reported adverse symptoms included tumor pain (*n* = 32, 17.7%), fever (*n* = 21, 11.6%), and fatigue (*n* = 16, 8.8%), while common grade III adverse events included febrile neutropenia (*n* = 13, 7.3%) and neutropenia (*n* = 9, 5.1%). There was no chemotherapy-related mortality (grade V) or anaphylaxis events.

## 1. Introduction

Primary bone malignancies are relatively rare worldwide in comparison to many other types of cancer [1]. They are classified into numerous subgroups, the most common of which are osteosarcoma and Ewing’s sarcoma [2]. From the previous report on the Asian population, the age-standardized incidence rate of osteosarcoma and Ewing’s sarcoma cases are approximately 3.01 and 0.52 per million person-years, respectively [3]. Patients with these bone sarcomas may present with many aspects of symptoms, including mass, wound, and pain. Curative therapy for both osteosarcoma and Ewing’s sarcoma requires a combination of surgical management and systemic chemotherapy [4]. With the multimodalities approach, the 5-year survival rate of osteosarcoma and Ewing’s sarcoma patients has been increased to approximately 60 to 70% [5,6]. However, patients with metastatic cancer still have a poor prognosis, with a 5-year survival rate lower than 25% [7,8,9].

Initially, osteosarcoma was resistant to several chemotherapy regimens, including those effective for Ewing’s sarcoma [10]. However, chemotherapy has become the principal treatment modality for osteosarcoma patients since the 1970s after the responses to Adriamycin (before the name “Doxorubicin” was introduced) [11] and high-dose Methotrexate [12] were observed. Recent advances in two- or three-drug combination chemotherapy, which later included Cisplatin and Ifosfamide, have improved the survival outcomes of osteosarcoma patients [13,14]. Several clinical trials have reported a 65–70% overall survival (OS) at 5 years for osteosarcoma patients being treated with Methotrexate plus Adriamycin plus Cisplatin (MAP) or Methotrexate plus Adriamycin plus Ifosfamide (MAI) [4,13,14,15]. From the existing evidence since the breakthrough of the chemotherapy era, however, adding any other chemotherapeutic agents to these three backbone drugs has failed to further improve the prognosis of osteosarcoma patients. Therefore, there is a high variation in chemotherapy regimens being used in osteosarcoma patients, particularly those who have poor responses to the standard chemotherapy regimen.

For Ewing’s sarcoma, the first and second Intergroup Ewing’s Sarcoma Studies (IESS) revealed a significant benefit of adding Doxorubicin to Cyclophosphamide plus Dactinomycin plus Vincristine, which was considered the conventional regimen for localized Ewing’s sarcoma in the 1990s [16,17]. Then, the combination of Vincristine plus Doxorubicin plus Cyclophosphamide plus Dactinomycin alternating with Ifosfamide plus Etoposide significantly improved 5-year disease-free survival (DFS) and OS rates at 70% for patients with localized disease in the landmark randomized controlled trial [18]. After that, this regimen has become the current standard chemotherapy regimen for Ewing’s sarcoma.

Chemotherapy can be administered as either neoadjuvant or adjuvant for osteosarcoma and Ewing’s sarcoma [19]. Neoadjuvant chemotherapy provides several advantages, including the reduction in local recurrence rates and the ability to histologically assess the response to chemotherapy by the tumor necrosis rate, one of the most reliable prognostic indicators [13,20,21]. Meanwhile, delaying definitive surgery owing to neoadjuvant chemotherapy does not significantly affect survival outcomes [13,22,23].

Although systemic chemotherapy is therapeutically effective in the treatment of bone sarcoma, it can damage the body’s environment and result in varying degrees of adverse drug reactions [24,25]. There have been many reports of adverse drug reactions from chemotherapy treatment in bone sarcomas [26]. The short-term complications may include infections associated with myelosuppression, mucositis, renal dysfunction, and hearing loss, while long-term complications, such as gonadal dysfunction and Doxorubicin-induced cardiac dysfunction, are not uncommon following chemotherapy treatment in bone sarcomas [27,28]. Some rare adverse drug reactions following chemotherapy treatment in bone sarcomas are sometimes reported, such as Doxorubicin-induced platelet procoagulant activity [29], and encephalopathy due to Cisplatin, Ifosfamide, or Etoposide [30,31,32]. It is observed that the more chemotherapeutic agents are included in chemotherapy regimens, the higher risk of adverse drug reactions following chemotherapy treatment increases [33,34,35]. For instance, the dose-limiting toxicities of Doxorubicin are myelosuppression and mucositis, and those of Cisplatin are nephrotoxicity and ototoxicity. The addition of Methotrexate to this drug combination (i.e., Doxorubicin and Cisplatin), particularly in patients with osteosarcoma, might worsen the drug’s adverse effects, which include mucositis and nephrotoxicity because of both overlapping toxicities and their own toxicities [36].

There is limited information on the longitudinal outcomes following chemotherapy treatment in patients with bone sarcoma, particularly in light of the experiences of cancer centers in developing countries. The objectives of this study were to evaluate the pattern of prescribed chemotherapy regimens for bone sarcomas, as well as the patients’ responses, survival outcomes, prognostic factors, and adverse events following chemotherapy treatment.

## 2. Materials and Methods

### 2.1. Study Design and Setting

This retrospective cohort study was conducted to evaluate prescribed chemotherapy regimens and responses, patient survival rates, and prognostic factors, as well as adverse outcomes in patients with osteosarcoma or Ewing’s sarcoma who had received chemotherapy treatment at the Maharaj Nakorn Chiang Mai Hospital, the referral cancer center in the North of Thailand, between 2008 and 2019. This article adheres to the Strengthening the Reporting of Observational Studies in Epidemiology (STROBE) recommendations [37].

### 2.2. Participants

Medical records of the patients with osteosarcoma or Ewing’s sarcoma who had received chemotherapy between 2008 and 2019 were retrieved using ICD-10 codes for the diagnosis of osteosarcoma or Ewing’s sarcoma [38]. Data were extracted from the electronic medical record at the Maharaj Nakorn Chiang Mai Hospital and the Chiang Mai Cancer Registry, which collected patient data in the form of scanned medical record sheets, of which the data included signs and symptoms, investigation and workup procedures, and the results, diagnosis, and management of the patients. This study included all patients who received at least one dose of chemotherapy treatment for either curative or palliative purposes; however, individuals whom we were unable to verify the definite diagnosis (by histopathology) or who were referred to other hospitals for chemotherapy treatment were excluded. Of a total of 324 cases diagnosed with bone and soft tissue sarcoma, 79 cases of osteosarcoma and 23 cases of Ewing’s sarcoma met the inclusion criteria for review and analysis of the study outcomes.

### 2.3. Variables

Patient characteristics at the time of diagnosis were extracted, including age, sex, and types of bone sarcoma based on a histopathological diagnosis. The adjudication of disease stage, which comprised localized (resectable or unresectable) tumor or metastatic tumor and tumor locations, was based on the consensus among oncologists, radiologists, and orthopedic oncologists, as well as findings from the patients’ investigations and workup procedures. We categorized the age group as <18 and ≥18 years for osteosarcoma and <25 and ≥25 years for Ewing’s sarcoma. The age categories were decided based on the evidence as reported elsewhere [39,40]. The patients were divided into two groups based on their staging status at diagnosis (localized or metastatic disease). In this study, patients with localized tumor(s) who underwent curative surgery were considered to have resectable disease, whereas patients with unresectable tumor(s) or metastasis at diagnosis or before undergoing curative surgery were considered to have unresectable/metastatic disease. Palliative surgery was defined as surgery done on patients who had an incurable or metastatic disease for palliative or non-curative purposes [41]. Information on previous radiation therapy, whether curative or palliative, was also gathered.

### 2.4. Primary Outcomes: Survival Outcomes and Prognostic Factors

Based on previous reports [42,43], the 5-year period for reporting survival outcomes was pre-determined. In patients with a resectable tumor(s), the time from the date of first chemotherapy treatment to the date of disease progression, as documented by investigation reports or oncologist’s records, was used to compute the 5-year DFS. In addition, we requested the survival data from the Chiang Mai Cancer Registry until the end of the follow-up period (17 June 2022) to yield the 5-year OS.

### 2.5. Secondary Outcomes: Prescribed Chemotherapy Regimens, Responses, and Adverse Effects

The initial regimen of chemotherapy was differentiated by its main purposes; the definition of each is as follows [44].

(1)Neoadjuvant chemotherapy: chemotherapy given to patients before surgical resection of the primary tumor. The response to neoadjuvant chemotherapy was determined by a tumor necrosis rate (TNR) from the post-operative histopathological report. According to the Huvos grading system [20], those with a TNR between 90% and 99% (grade III) or 100% (grade IV) were considered responders, whereas those with a TNR of less than 50% (grade I) or between 50% and 89% (grade II) were considered non-responders.(2)Adjuvant chemotherapy: chemotherapy given to patients after surgical resection of the primary tumor to kill any remaining cancer cells with the aim of reducing the risk of tumor recurrence.(3)Palliative chemotherapy: chemotherapy given to patients who presented with unresectable/metastatic disease in order to relieve symptoms and lessen cancer-related suffering or in the event of disease progression or relapse after receiving multimodal treatment.

For adverse outcome assessments, patients’ self-reported symptoms were collected, and adverse events following chemotherapy treatment were assessed using the Common Terminology Criteria for Adverse Events (CTCAE) criteria, version 5.0 [45].

### 2.6. Treatment Strategy

For the treatment strategy for our patients with localized osteosarcoma and Ewing’s sarcoma, surgical resection of the primary tumor with adequate margins was an essential component of the curative strategy for patients with localized disease. Therefore, the chemotherapy for both osteosarcoma and Ewing’s sarcoma might start before (neoadjuvant) or after (adjuvant) curative surgical resection of the primary tumor. In our study, the patterns of chemotherapy prescription mainly depended on the schedule for surgery, the adequacy of primary surgical margins, chemotherapy adverse effects, and the progression of disease following the initial chemotherapy treatment. Generally, almost all patients would receive neoadjuvant chemotherapy as their initial treatment or undergo curative surgery if it was available and not delayed by waiting to complete neoadjuvant chemotherapy treatments. If patients had completed neoadjuvant chemotherapy and definitive surgery with adequate surgical margins, adjuvant chemotherapy usually would not be given after curative surgery. On the other hand, patients would be given adjuvant chemotherapy if they had not finished neoadjuvant chemotherapy before having curative surgery or if they had finished neoadjuvant chemotherapy, but the surgical margins were not adequate. The adjuvant chemotherapy regimen would continue with the same regimen or a new regimen that included the same drugs used in the initial neoadjuvant chemotherapy unless there were adverse effects from the chemotherapy or the progression of the disease during treatment.

For an unresectable tumor or metastasis, preoperative chemotherapy followed by surgery to remove the primary tumor and surgical resection of bone metastatic lesions followed by postoperative combination chemotherapy were employed as the primary approach. The alternative approach, starting with surgery for the primary tumor, followed by chemotherapy, and then surgical resection of metastatic bone lesions, would be employed if the primary approach was not appropriate (i.e., in patients with intractable pain, pathological fracture, or uncontrolled infection of the tumor, which could increase the risk of sepsis).

For radiation therapy, the objective of this treatment was local control. Radiation therapy was employed in the following cases: (1) localized resectable disease for which surgical margins could not be evaluated or at an inadequate surgical margin site; (2) unresectable disease; and (3) metastatic bone lesions.

### 2.7. Statistical Analysis

Descriptive statistics were used to describe patient characteristics, prescribed chemotherapy regimens, responses to chemotherapy, and adverse effects following chemotherapy treatment. Categorical data are presented as a frequency distribution with percentages. Continuous data are presented as mean with standard deviation (SD) or median with interquartile range (IQR), as appropriate. The flexible parametric survival model with two-knot restricted cubic splines was performed to investigate the prognostic factors for 5-year DFS and 5-year OS. Survival curves with adjusted hazard ratios (aHR) and their 95% confidence intervals (CI) were used to reflect the risk of 5-year DFS and 5-year OS. To quantify the effect of potential prognostic factors as unbiasedly and precisely as possible, the potential prognostic factors were chosen based on the ground theory and the established evidence, including age, sex, radiation therapy, disease status, and post-neoadjuvant chemotherapy response (for osteosarcoma). The survival model for osteosarcoma was also adjusted for the time-varying effect of post-neoadjuvant chemotherapy response. Statistical analyses were conducted using the STATA statistical software program, version 16 [46]. All statistical analyses were two-sided, and a *p*-value of 0.05 was considered to indicate statistical significance.

## 3. Results

### 3.1. Patient Characteristics

A total of 102 patients were eligible for inclusion. All of the participants were Asian. The patient characteristics are shown in Table 1.

#### 3.1.1. Osteosarcoma

Of the 79 osteosarcoma patients, 41 (51.9%) were male, and 48 (60.8%) were 18 years or older. The most common location of the primary tumor was the extremities (78.5%). At the time of diagnosis, most osteosarcoma patients (91.1%) had a localized, 77.8% of which were resectable. Fifteen osteosarcoma patients with resectable disease received radiation therapy after definitive surgery because surgical margins were inadequate or could not be evaluated. From a total of 23 osteosarcoma patients who had unresectable disease or metastasis, 15 patients underwent palliative surgery to remove primary tumors or metastatic bone lesions for local control, and eight patients received radiation therapy to the extremities for local control at an unresectable tumor or metastatic bone lesions or inadequate surgical margins site.

#### 3.1.2. Ewing’s Sarcoma

Of the 23 osteosarcoma patients, 15 (65.2%) were male, and 18 (78.3%) were 18 years or older. The most common location of the primary tumor was the extremities (39.1%). At the time of diagnosis, around three-fourths of Ewing’s sarcoma patients (78.3%) had a localized tumor at diagnosis, 50% of which were resectable. Three Ewing’s sarcoma patients with resectable disease received radiation therapy after definitive surgery because surgical margins could not be evaluated. From nine Ewing’s patients who had unresectable disease or metastasis, two patients underwent palliative surgery to remove primary tumors or metastatic bone lesions for local control, and five patients received radiation therapy to the extremities for local control at an unresectable tumor or metastatic bone lesions or inadequate surgical margins site.

### 3.2. Survival Outcomes and Prognostic Factors

In this study, the median follow-up time was 27 (IQR, 5 to 92) months since the date of the first chemotherapy treatment.

#### 3.2.1. Osteosarcoma

At the end of the follow-up period, 41 (51.9%) osteosarcoma patients were dead. The 5-year DFS and 5-year OS rates were 39.6% (disease progression 42, lost 19) and 45.5% (death 39, lost 15). The aHR and survival probabilities of 5-year DFS and 5-year OS using a flexible parametric survival analysis for osteosarcoma are illustrated in Figure 1a–j. A status of having a localized resectable tumor(s) was found to be significantly associated with better 5-year DFS (aHR 0.17, 95% CI 0.07–0.40, *p* < 0.001) and 5-year OS (aHR 0.21, 95% CI 0.09–0.50, *p* < 0.001). The estimated 5-year DFS and 5-year OS in patients with resectable disease were 60.9% and 63.3%, whereas the 5-year DFS and 5-year OS for those with unresectable/metastatic disease were 21.3% and 23.3%, respectively. Patients with a good response to neoadjuvant chemotherapy (TNR > 90%) also had a significantly better prognosis for both 5-year DFS (aHR 0.08, 95% CI 0.01–0.62, *p* = 0.016) and 5-year OS (aHR 0.09, 95% CI 0.01–0.74, *p* = 0.024), and so did female patients in both 5-year DFS (aHR 0.33, 95% CI 0.16–0.72, *p* = 0.005) and 5-year OS (aHR 0.41, 95% CI 0.20–0.87, *p* = 0.019). Other factors, including age and radiation therapy, were not significantly associated with the survival outcomes (Appendix A).

#### 3.2.2. Ewing’s Sarcoma

At the end of the follow-up period, 12 (52.2%) Ewing’s sarcoma patients were dead. The 5-year DFS and 5-year OS rates were 40.9% (disease progression 13, lost 2) and 45.5% (death 12, lost 2). The aHR and survival probabilities of 5-year DFS and 5-year OS using a flexible parametric survival analysis for Ewing’s sarcoma are illustrated in Figure 2a–h. The estimated 5-year DFS and 5-year OS in patients with resectable disease were 54.4% and 88.3%, while the 5-year DFS and 5-year OS for those with unresectable/metastatic disease were 12.3% and 18.4%, respectively. A status of having a localized resectable tumor(s) was found to have better 5-year DFS (aHR 0.16, 95% CI 0.04–0.67, *p* = 0.012) and 5-year OS (aHR 0.06, 95% CI 0.01–0.51, *p* = 0.010). We also found that adult patients (≥25 years) had significantly poorer 5-year DFS (aHR 6.05, 95% CI 1.46–25.10, *p* = 0.013) and 5-year OS (aHR 6.79, 95% CI 1.33–34.72, *p* = 0.021). The association between the survival outcomes and other factors consisting of sex and radiation therapy was not observed (Appendix A).

### 3.3. Chemotherapy Prescribing Patterns and Responses

The chemotherapy prescribing patterns for osteosarcoma and Ewing’s sarcoma are shown in Table 2.

#### 3.3.1. Osteosarcoma

For osteosarcoma, Cisplatin plus Doxorubicin was the most used regimen in both neoadjuvant (*n* = 30/41, 73.2%) and adjuvant therapy (*n* = 20/47, 42.6%). Of 72 osteosarcoma patients with localized disease at diagnosis, 32 patients (44.4%) received neoadjuvant chemotherapy and were assessed for post-neoadjuvant chemotherapy response (Table 3). Around one-fifth of them (*n* = 7/32, 21.9%) were considered responders (TNR ≥ 90%). The chemotherapy regimens commonly used following disease progression were Cisplatin plus Doxorubicin (*n* = 13/32, 40.6%), followed by Ifosfamide plus Etoposide (*n* = 5/32, 15.6%).

#### 3.3.2. Ewing’s Sarcoma

For Ewing’s sarcoma, the standard regimen (Vincristine plus Doxorubicin plus Cyclophosphamide alternating with Ifosfamide plus Etoposide) was commonly prescribed for neoadjuvant (*n* = 4/9, 44.4%) and adjuvant therapy (*n* = 2/5, 40.0%), followed by Ifosfamide plus Etoposide (*n* = 3/9, 33.3%) for neoadjuvant therapy and Vincristine plus Doxorubicin plus Cyclophosphamide for adjuvant therapy (*n* = 2/5, 40.0%). Two-thirds of the patients with resectable disease (*n* = 6/9, 66.6%) received neoadjuvant therapy, with only three of them having an available report assessing post-neoadjuvant treatment response. None of them were considered a responder. The regimens commonly used following disease progression were Ifosfamide plus Etoposide (*n* = 9/24, 37.5%) and Cisplatin plus Etoposide (*n* = 6/24, 25.0%).

### 3.4. Self-Reported Symptoms and Adverse Events

Due to the fact that there were a variety of prescribed regimens in our study and that each cycle typically included a specific combination of chemotherapy that can cause different or overlapping adverse effects, it was difficult to determine which single agent was the causative agent, especially for common symptoms and adverse events. In addition, an increase in the number of specific chemotherapy drugs in each cycle may improve clinical outcomes while increasing the risk of adverse effects. Thus, the self-reported adverse symptoms and adverse events are summarized by the number of specific chemotherapy drugs in each cycle in Table 4 and Table 5, respectively. Recorded data showed that 181 chemotherapy cycles resulted in self-reported symptoms by the patients who received them. Of the 181 cycles provided, common self-reported adverse symptoms included tumor pain (32 cycles, 17.7%), fever (21 cycles, 11.6%), and fatigue (16 cycles, 8.8%). Common grade III adverse events included febrile neutropenia (13 cycles, 7.3%) and neutropenia (9 cycles, 5.1%). Other grade II–III adverse events included unspecified infections (*n* = 23, 12.7%), while acute kidney injury, mucositis, oral ulcers, and other hematologic toxicities were observed in less than 1%. One Ewing’s sarcoma patient who received Ifosfamide plus Etoposide had grade III encephalopathy, and one osteosarcoma patient who received Ifosfamide plus Doxorubicin had grade IV venous sinus thrombosis. There was no chemotherapy-related mortality (grade V) or anaphylaxis events.

## 4. Discussion

Although chemotherapy is one of the most effective modalities for bone sarcomas, there have been a few reports on the experience of such treatment modalities in developing countries [47,48,49,50]. For osteosarcoma, the most common regimen for neoadjuvant, adjuvant, and palliative chemotherapy in our setting was Cisplatin plus Doxorubicin, followed by Ifosfamide plus Doxorubicin, and its combination with Cisplatin or Methotrexate. Cisplatin and Doxorubicin are one of the most common chemotherapy agents, with well-established efficacy in osteosarcoma by several clinical studies [51,52,53,54]. Although high-dose Methotrexate is a standard agent in many countries in the combination regimen, especially in pediatrics with osteosarcoma, this agent was uncommonly used in very few of our adult patients due to the not being readily available for drug-level concentration monitoring for routine practice and concerns over toxicity, and some skepticism regarding efficacy when adding to the commonly used doublet cisplatin and doxorubicin. Methotrexate could increase the risk of some complications (e.g., nephrotoxicity, mucositis, and myelosuppression) when the drug is combined with Cisplatin [55], Doxorubicin [28,56], or Ifosfamide [57,58]. In addition, Methotrexate clearance in adults is known to be slower and less predictable than that in pediatric patients [59,60]. This could be explained why the two-drug combination was commonly used instead of the three-drug combination. Ifosfamide was also one of the common agents being used in our setting. From the large clinical study of the Children’s Oncology Group (COG), Ifosfamide is found to be as effective as Cisplatin when it is combined with Methotrexate plus Doxorubicin in terms of the response rate as represented by post-neoadjuvant TNR. This may be the reason why Ifosfamide plus Doxorubicin was the second most prevalent combination in our study, with a response rate of 66.7%.

For Ewing’s sarcoma, the standard regimen, consisting of Vincristine plus Doxorubicin plus Cyclophosphamide alternating with Ifosfamide plus Etoposide, was the most commonly used regimen in this study. The standard regimen showed a slightly lower response at 70% in our study when compared to a previous study with a response rate of around 90% [61]. Two patients who were scheduled to receive the standard regimen for neoadjuvant chemotherapy but rather received only Ifosfamide and Etoposide demonstrated a poor response (TNR < 50%).

Our study revealed that a response to neoadjuvant chemotherapy (TNR > 90%) was significantly associated with better 5-year DFS and 5-year OS in osteosarcoma patients, and the adjusted estimated 5-year DFS and 5-year OS were substantially high at approximately 90% when compared to previous reports [4,13,14,15]. Since Huvos et al. [20,62] first described the histologic evidence in bone sarcoma patients treated with neoadjuvant chemotherapy, the TNR after neoadjuvant chemotherapy has become one of the most important prognostic factors for bone sarcomas. A TNR of greater than 90% has been established as a favorable indicator related to improved 5-year DFS and 5-year OS through multiple clinical studies [20,63,64]. Although Alaya et al. [65] suggested that a TNR of greater than 60% would also indicate a definite chemotherapeutic effect, there is no significant difference in 5-year DFS and 5-year OS between osteosarcoma patients with the TNR between 50 and 89% after neoadjuvant chemotherapy and those with adjuvant treatment only. Our observation was consistent with several clinical studies, suggesting that a response to neoadjuvant chemotherapy, as defined by the TNR of greater than 90%, is a significant prognostic factor of survival outcomes irrespective of adjuvant chemotherapy regimens [13,15,66].

It Is well established that the detection of localized resectable disease in the early stage of bone sarcoma is related to longer survival times [67]. Since the development of a multimodal treatment strategy, the survival outcome of patients with resectable disease has been substantially improved; however, that of patients with unresectable/metastatic disease still remains poor [68,69]. In accordance with several published studies [63,70,71,72,73], the patients with resectable osteosarcoma or Ewing’s sarcoma in our study also showed significantly higher 5-year DFS and 5-year OS when compared to their counterparts with unresectable/metastatic disease. The estimated 5-year DFS and 5-year OS for resectable osteosarcoma were observed to be around 60% in this study, which is rather comparable to a previous study [74]. For Ewing’s sarcoma patients with resectable disease, the estimated 5-year OS was 88.3%, which is slightly higher than that of 70–80% reported in other studies [4,18,73,75]. For patients with unresectable/metastatic disease, the estimated 5-year DFS and 5-year OS remained below 25% in both osteosarcoma and Ewing’s sarcoma, and our observation is similar to those reported in other studies [73,74].

We also found that, among osteosarcoma patients, the female gender was significantly associated with longer 5-year DFS and 5-year OS. Our finding is consistent with a recent meta-analysis of prognostic factors in osteosarcoma patients [63]. It is previously observed that male patients with osteosarcoma tend to have a lower response to neoadjuvant chemotherapy when compared to female patients [5]. The differences in survival outcomes between males and females may be explained by the way endogenous sex hormones affect osteosarcoma cells. In the androgen receptor coactivator-knockout model, the absence of the androgen receptor signaling pathway could inhibit proliferation-related signaling and subsequently decrease osteosarcoma cell proliferation [76]. A high dose of 17β-estradiol treatment is previously shown to suppress the proliferation, migration, and invasion processes of osteosarcoma cells [77]. According to the existing evidence, patients in the older adolescent and young adult age groups [39,78], generally referred to as those between the ages of 18 and 40 years, tend to have a worse prognosis. Because the age group was reported to have a minor effect on survival outcomes when compared to other prognostic factors, the effect of the age group in our study might not be highlighted due to a relatively small sample size. For Ewing’s sarcoma, patients who were 25 years of age or older in our study showed significantly lower 5-year DFS and 5-year OS. Our finding was consistent with the previous findings on different age groups of the patients, indicating that adolescents and adults with Ewing’s sarcoma had a worse prognosis than younger individuals [18,40,79,80,81]. The association between older age and survival outcomes in Ewing’s sarcoma could be explained by the fact that adults with Ewing’s sarcoma often had metastatic disease at diagnosis, unfavorable sites of tumors, a higher tumor volume, and poor clinical outcomes after treatments [18,40].

The observation of some adverse outcomes warrants further discussion. In our cohort, tumor pain was the most frequent self-reported adverse symptom after chemotherapy. However, the literature suggests that this symptom is likely to be attributable to disease conditions rather than chemotherapy [16,17,53]. Grade III febrile neutropenia or neutropenia was reported to be around 12.5% of the total chemotherapy prescriptions. This incidence is consistent with previous reports, with the incidence of grade III febrile neutropenia ranging from less than 1 to 24% [14,15,66,82,83]. Other grade II–III adverse events in our study included unspecified infections, nephrotoxicity, other hematologic toxicities, and mucositis, all of which were reported with a lower incidence than that observed in other previous studies [14,15,66,83]. Chemotherapy-induced encephalopathy (grade III) was observed in two Ewing’s sarcoma patients who received Ifosfamide or Ifosfamide plus Etoposide. Ifosfamide is regarded to be the most probable cause of encephalopathy due to the neurotoxicity of its active metabolite, chloroacetaldehyde [84]. Chloroacetaldehyde can cross the blood-brain barrier and can inhibit flavoproteins, the mitochondrial respiratory chain, and the oxidation of NADH by thialysine ketimine [85]. Etoposide could also be another possible cause of this complication, as it is found to be associated with posterior reversible encephalopathy syndrome [86]. During a period of 12 years of experience in our setting, there was only one life-threatening adverse event (grade IV), which is venous sinus thrombosis, which was observed in one osteosarcoma patient who received Ifosfamide plus Doxorubicin. Doxorubicin could be the likely cause of this serious adverse event as it has the ability to enhance platelet hyperreactivity through induced thrombin generation [29]. In our study, we did not observe gonadal dysfunction or Doxorubicin-induced cardiomyopathy, which is a long-term, dose-dependent, fatal, but uncommon adverse event following chemotherapy treatment for bone sarcomas [14,15,66,83,87,88].

In this study, several limitations should be taken into account. First, some data might not have been documented in the electronic medical record unless the physicians had recognized its importance or relevance. Due to the retrospective nature of the study, there were several missing data on TNRs following neoadjuvant chemotherapy, especially in patients with Ewing’s sarcoma. It is also possible that the self-reported adverse symptoms and adverse events might be underestimated. Second, it is not feasible at the moment, with the small sample size and the heterogeneity of prescribed chemotherapy regimens, to compare the survival outcomes and adverse events across various chemotherapy regimens. Finally, because the reported descriptive data could be under-detected due to the small sample size, the comparison of our patients’ characteristics, prescribed chemotherapy patterns, and chemotherapy responses to other studies, particularly for Ewing’s sarcoma, should be done with caution and awareness of this limitation. Since bone sarcomas are rare disease, it may be challenging to collect sufficient sample sizes at just one center.

## 5. Conclusions

There was high variability in chemotherapy regimens being used in osteosarcoma patients in our setting. Two-drug regimens, i.e., Cisplatin plus Doxorubicin and Ifosfamide plus Doxorubicin, were the most commonly used regimens for osteosarcoma. In contrast, the variation in chemotherapy regimen for Ewing’s sarcoma (i.e., Vincristine plus Doxorubicin plus Cyclophosphamide alternating with Ifosfamide plus Etoposide). In our study, the 5-year DFS and 5-year OS of the patients with localized resectable disease were comparable to those reported in other previous studies conducted at other experienced cancer centers, whereas the survival outcomes of the patients with unresectable/metastatic disease remained poor. Response to neoadjuvant chemotherapy and female gender were independent favorable prognostic factors for osteosarcoma. Adult patients with Ewing’s sarcoma were significantly associated with poor survival outcomes. Common adverse outcomes following chemotherapy included tumor pain, fever, fatigue, and febrile neutropenia.

## Figures and Tables

**Figure 1 cancers-15-01979-f001:**
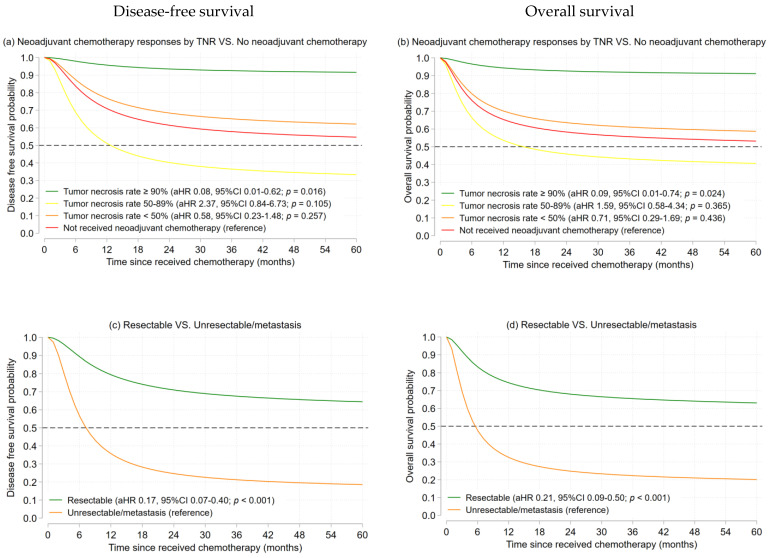
The estimated 5-year DFS and 5-year OS of osteosarcoma patients by the Flexible parametric survival model (**a**–**j**).

**Figure 2 cancers-15-01979-f002:**
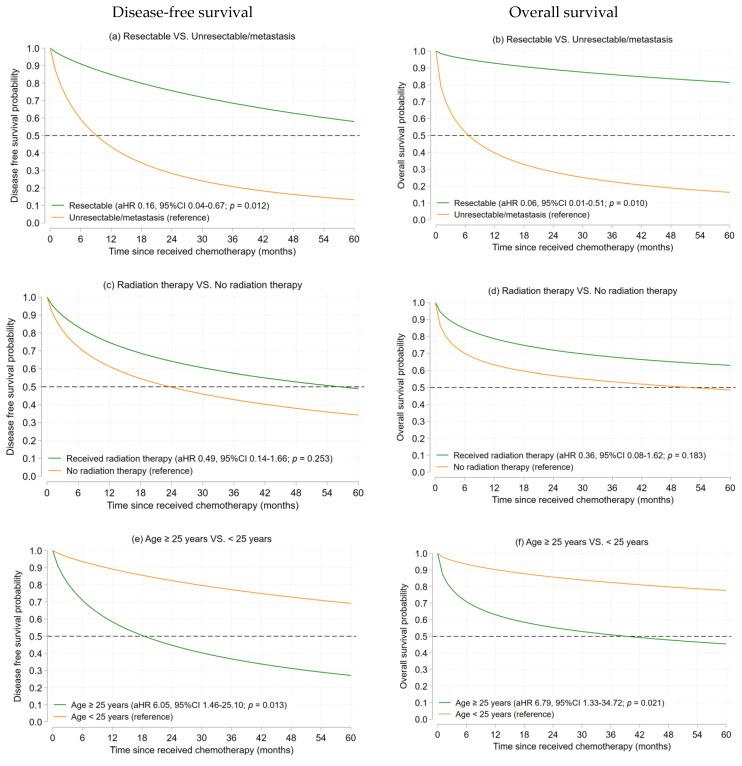
The estimated 5-year DFS and 5-year OS of Ewing’s sarcoma patients by the Flexible parametric survival model (**a**–**h**).

**Table 1 cancers-15-01979-t001:** Patient characteristics.

Characteristics	Osteosarcoma (*n* = 79)		Ewing’s Sarcoma (*n* = 23)	
	*n*	%	*n*	%
**Sex**				
Male	41	(51.9)	15	(65.2)
Female	38	(48.1)	8	(34.8)
Median age, year (IQR)	20	(16–48)	23	(18–30)
Location of the primary tumor				
Head and neck	3	(3.8)	2	(8.7)
Trunk	9	(11.4)	6	(26.1)
Intraperitoneal	2	(2.5)	3	(13.0)
Extremities	62	(78.5)	9	(39.1)
Others	3	(3.8)	3	(13.1)
Localized tumor at diagnosis	72	(91.1)	18	(78.3)
Resectable tumor with curative surgery	56	(77.8)	9	(50.0)
Unresectable tumor	4	(5.5)	6	(33.3)
Metastasis after prior treatment	12	(16.7)	3	(16.7)
Metastasis at diagnosis	7	(8.9)	5	(21.7)
Palliative surgery	15	(78.9)	2	(25.0)
Radiation therapy	23	(29.1)	8	(34.8)
Chemotherapy for resectable disease				
Neoadjuvant only	21	(26.6)	7	(30.4)
Adjuvant only	27	(34.2)	3	(13.0)
Neoadjuvant followed by adjuvant	20	(25.3)	2	(8.7)
Chemotherapy for unresectable/metastatic disease				
First-line treatment	20	(25.3)	12	(52.2)
Second-line treatment	11	(13.9)	7	(30.4)
Third-line treatment	1	(1.3)	5	(21.7)

**Table 2 cancers-15-01979-t002:** The chemotherapy prescribing patterns for osteosarcoma and Ewing’s sarcoma.

Chemotherapy Regimens	Osteosarcoma (*n* = 79)		Chemotherapy Regimens	Ewing’s Sarcoma (*n* = 23)	
	*n*	(%)		*n*	(%)
**Neoadjuvant only**	**21**		**Neoadjuvant only**	**7**	
AP	17	(81.0)	IE	3	(42.8)
AI	2	(9.5)	VAC + IE	2	(28.6)
A	2	(9.5)	VAC	1	(14.3)
			AI	1	(14.3)
**Adjuvant only**	**27**		**Adjuvant only**	**3**	
AP	13	(48.2)	VAC + IE	2	(66.7)
AI	4	(14.8)	PE	1	(33.3)
IE	3	(11.1)			
AP + IE	2	(7.4)			
MAP	1	(3.7)			
P + IE	1	(3.7)			
MP + IE	1	(3.7)			
MAP + IE	1	(3.7)			
PIE + Cy	1	(3.7)			
**Neoadjuvant followed by adjuvant**	**20**		**Neoadjuvant followed by adjuvant**	**2**	
AP/AP	5	(25.0)	VAC + IE/VAC	2	(100.0)
AP/IE	4	(20.0)			
MAP/AP	2	(10.0)			
AP/P + IE	2	(10.0)			
AP/AI	1	(5.0)			
AP/AP + IE	1	(5.0)			
MAP/M	1	(5.0)			
AP + IE/G + D	1	(5.0)			
AI/AI	1	(5.0)			
AI/M	1	(5.0)			
Ir + P/I	1	(5.0)			
**Chemotherapy regimen for unresectable** **/** **metastatic disease**					
**First** **-line**	**20**		**First** **-line**	**12**	
AP	11	(55.0)	PE	6	(50.0)
IE	4	(20.0)	IE	4	(33.4)
I	3	(15.0)	VAC	1	(8.3)
AI	1	(5.0)	A	1	(8.3)
Cy + E	1	(5.0)			
**Second** **-line**	**11**		**Second** **-line**	**7**	
I	3	(27.3)	IE	3	(42.8)
G	3	(27.3)	VAC	2	(28.6)
AP	2	(18.1)	V + IE	1	(14.3)
PE	1	(9.1)	I	1	(14.3)
IE	1	(9.1)			
G + DTX	1	(9.1)			
**Third** **-line**	**1**		**Third** **-line**	**5**	
Tr	1	(100.0)	IE	2	(40.0)
			I	2	(40.0)
			AP	1	(20.0)

Abbreviations: A, Doxorubicin; C, Cyclophosphamide; DTX, Docetaxel; E, Etoposide; G, Gemcitabine; I, Ifosfamide; Ir, Irinotecan; M, Methotrexate; P, Cisplatin, Tr, Trabectedin; V, Vincristine.

**Table 3 cancers-15-01979-t003:** Response to neoadjuvant chemotherapy in osteosarcoma patients with a localized resectable tumor(s).

Chemotherapy Regimens	(*n* = 32) *n* (%)	Tumor Necrosis Rate after Neoadjuvant Chemotherapy, *n* (%)		
		<50%	50–89%	≥90%
AP	25 (78.1)	15 (60.0)	7 (28.0)	3 (12.0)
AI	3 (9.4)		1 (33.3)	2 (66.7)
AP + IE	2 (6.3)	1 (50.0)	1 (50.0)	
MAP	1 (3.1)			1 (100.0)
A	1 (3.1)			1 (100.0)

Abbreviations: A, Doxorubicin; E, Etoposide; I, Ifosfamide; M, Methotrexate; P, Cisplatin.

**Table 4 cancers-15-01979-t004:** Self-reported adverse symptoms following chemotherapy treatment.

Self-Reported Adverse Symptoms		Osteosarcoma			Ewing’s Sarcoma		
		1 Drug (*n* = 22)	2 Drugs (*n* = 105)	3 Drugs (*n* = 7)	1 Drug (*n* = 4)	2 Drugs (*n* = 27)	3 Drugs (*n* = 16)
Alteration of consciousness		-	-	-	1 (25.0%)	-	-
Anorexia		-	1 (1.0%)	-	-	-	-
Diarrhea		-	-	-	1 (25.0%)	-	-
Dyspnea		2 (9.1%)	6 (5.7%)	2 (28.6%)	-	1 (3.7%)	-
Fatigue		3 (13.6%)	10 (9.5%)	-	-	3 (11.1%)	-
Fever		3 (13.6%)	5 (4.8%)	4 (57.1%)	-	4 (14.8%)	5 (31.3%)
Nausea		2 (9.1%)	4 (3.8%)	-	-	2 (7.4%)	1 (6.3%)
Oral ulcer		-	1 (1.0%)	-	-	-	-
Rash		-	-	1 (14.3%)	-	-	-
Tumor pain		5 (22.7%)	21 (20.0%)	2 (28.6%)	-	1 (3.7%)	3 (18.8%)
Weakness		-	-	-	-	3 (11.1%)	-
Wound		1 (4.5%)	2 (1.9%)	-	-	1 (3.7%)	-

The term “drug” refers to the number of specific chemotherapy drugs in each cycle. *n*, number of cycle.

**Table 5 cancers-15-01979-t005:** Adverse drug reactions following chemotherapy treatment.

Adverse Drug Reaction with Grade II–IV	Grade	Osteosarcoma			Ewing’s Sarcoma		
		1 Drug (*n* = 22)	2 Drugs (*n* = 105)	3 Drugs (*n* = 7)	1 Drug (*n* = 4)	2 Drugs (*n* = 27)	3 Drugs (*n* = 16)
Vascular and hematologic system							
Anemia	II	-	1 (0.9%)	-	1 (25.0%)	-	-
	III	1 (4.5%)	-	-	-	-	-
Neutropenia	III	-	5 (4.8%)	-	-	3 (11.1%)	1 (6.3%)
Thrombocytopenia	III	-	1 (0.9%)	-	-	-	-
Pancytopenia	III	-	2 (1.9%)	-	-	1 (3.7%)	-
Thrombosis	III	-	1 (0.9%)	-	-	-	-
	IV	-	1 (0.9%)	-	-	-	-
Infection							
Febrile neutropenia	III	-	6 (5.7%)	1 (14.3%)	-	2 (7.4%)	4 (25.0%)
Unspecified infection	II	2 (9.1%)	10 (9.5%)	3 (42.9%)	1 (25.0%)	3 (11.1%)	3 (18.8%)
	III	-	1 (0.9%)	-	-	-	-
Nephrology							
Acute kidney injury	III	-	-	-	-	1 (3.7%)	-
Hypokalemia	III	-	1 (0.9%)	-	-	-	-
Neuropsychology							
Encephalitis	II	-	-	-	1 (25.0%)	-	-
	III	-	-	-	-	1 (3.7%)	-
Gastrointestinal system							
Mucositis	II	-	1 (0.9%)	-	-	-	-
Peptic ulcer	II	-	1 (0.9%)	-	-	-	-
Integument							
Oral ulcer	II	-	1 (0.9%)	-	-	-	-

The term “drug” refers to the number of specific chemotherapy drugs in each cycle. *n*, number of cycle.

## Data Availability

The data underlying this article will be shared on reasonable request to the corresponding author.

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
