# Peer review of "The Survival Outcomes, Prognostic Factors and Adverse Events following Systemic Chemotherapy Treatment in Bone Sarcomas: A Retrospective Observational Study from the Experience of the Cancer Referral Center in Northern Thailand"

_cancers, 2023, doi:10.3390/cancers15071979_

Round 1
Reviewer 1 Report
Thank you for the opportunity to review this work. The authors present a retrospective study performed on patients with Osteosarcoma and Ewing sarcoma undergoing therapy at a reference center in the North of Thailand between 2008 and 2019. The aim of this study was to evaluate the pattern of prescribed chemotherapy (CT) regimens, patient survival rates, prognostic factors and adverse events following treatments.
Comments:
- the authors combine osteosarcoma and Ewing's sarcoma with two completely different pathways from the point of view of therapy, pathology ... This creates confusion in reading the data (why not present it separately?)
- the choice to do very few cycles of MTX in osteosarcoma (given the data published by Ferrari and Bielack) was made for the age of the patients? A more detailed comment is needed.
- The choice to perform 23 RT and 15 palliative surgery on 79 patients with osteosarcoma with only 4 unresectable tumor needs to be better explained
- It is necessary to explain better why the choice of neoadjuvant only CT or adjuvant only CT was made in patients with resectable osteosarcoma when the data in the literature confirm the importance of neoadjuvant chemotherapy followed by adjuvant. The same thing must also be explained for Ewing's sarcoma.
- with such important biases from the chemotherapy point of view and with such a small sample size, I find it very difficult to highlight prognostic factors.
Author Response
Point 1: Thank you for the opportunity to review this work. The authors present a retrospective study performed on patients with Osteosarcoma and Ewing sarcoma undergoing therapy at a reference center in the North of Thailand between 2008 and 2019. The aim of this study was to evaluate the pattern of prescribed chemotherapy (CT) regimens, patient survival rates, prognostic factors and adverse events following treatments.
Response 1: We would like to thank the reviewer for the suggestion and encouragement.
Point 2: The authors combine osteosarcoma and Ewing's sarcoma with two completely different pathways from the point of view of therapy, pathology ... This creates confusion in reading the data (why not present it separately?)
Response 2: We agree with the reviewer’s point about the different perspectives on osteosarcoma and Ewing's sarcoma that could confuse readers. We have revised the main texts in the result sections by separately describing the characteristics and other information of osteosarcoma and Ewing’s sarcoma in the different subsections as shown below. However, a constraint on a table and a figure limit in the journal, as well as the quantity of information, prevent us from presenting the results in table format separately.
- Results (with subsections of osteosarcoma and Ewing’s sarcoma in each section)
3.1. Patient Characteristics
3.1.1 Osteosarcoma
3.1.2 Ewing’s sarcoma
3.2. Survival Outcomes and Prognostic Factors
3.2.1 Osteosarcoma
3.2.2 Ewing’s sarcoma
3.3. Chemotherapy Prescribing Patterns and Responses
3.3.1 Osteosarcoma
3.3.2 Ewing’s sarcoma
3.4. Self-reported Symptoms and Adverse Events
Moreover, there are common points of view on their epidemiology and treatment strategy. Osteosarcoma and Ewing's sarcoma are two of the most common primary bone tumors in children and adolescents, and their treatment landscape usually involves surgery, radiation therapy, and multiple chemotherapy agents. Because both osteosarcoma and Ewing’s sarcoma care teams in our context and most in developing countries involve the same patient care team (i.e., oncologists, radiation oncologists, and orthopedic oncologists), this information could be useful for their practice, particularly for other bone cancers care team in developing countries, where there is limited information on the reported longitudinal treatment outcomes.
Point 3: The choice to do very few cycles of MTX in osteosarcoma (given the data published by Ferrari and Bielack) was made for the age of the patients? A more detailed comment is needed.
Response 3: We have revised and added further discussion. It now reads “Although high-dose Methotrexate is a standard agent in many countries in the combination regimen, especially in pediatrics with osteosarcoma, this agent was uncommonly used in very few of our adult patients due to the not being readily available for drug-level concentration monitoring for routine practice and concerns over toxicity, and some skepticism regarding efficacy when adding to the commonly used doublet cis-platin and doxorubicin. Methotrexate could increase the risk of some complications (e.g., nephrotoxicity, mucositis, and myelosuppression) when the drug is combined with Cisplatin [54], Doxorubicin [28,55], or Ifosfamide [56,57]. In addition, Methotrexate clearance in adults is known to be slower and less predictable than that in pediatric patients [58,59]. This could be explained why the two-drug combination was commonly used instead of the three-drug combination.” (Page 13, Line 356-367).
Point 4: The choice to perform 23 RT and 15 palliative surgery on 79 patients with osteosarcoma with only 4 unresectable tumor needs to be better explained
Response 4: We would like to thank the reviewer to point out the unclear issues in our results. We would like to clarify that there were not only four osteosarcoma patients with unresectable disease at diagnosis but also seven patients with metastasis at diagnosis and 12 patients who found evidence of metastasis after receiving neoadjuvant chemotherapy as described in Table 1. For more clarity, we have added further explanation in section 3.1. patients’ characteristics.". It now reads "Fifteen osteosarcoma patients with a resectable disease received radiation therapy after definitive surgery because surgical margins were inadequate or could not be evaluated. From a total of 23 osteosarcoma patients who had an unresectable disease or metastasis, 15 patients underwent palliative surgery to remove primary tumors or metastatic bone lesions for local control, and 8 patients received radiation therapy to the extremities for local control at an unresectable tumor or metastatic bone lesions or inadequate surgical margins site." (Page 5, Line 240-246) and "Three Ewing’s sarcoma patients with a resectable disease received radiation therapy after definitive surgery because surgical margins could not be evaluated. From nine Ewing’s patients who had unresectable disease or metastasis, two patients underwent palliative surgery to remove primary tumors or metastatic bone lesions for local control, and five patients received radiation therapy to the extremities for local control at an unresectable tumor or metastatic bone lesions or inadequate surgical margins site." (Page 6, Line 251-257)
Point 5: It is necessary to explain better why the choice of neoadjuvant only CT or adjuvant only CT was made in patients with resectable osteosarcoma when the data in the literature confirm the importance of neoadjuvant chemotherapy followed by adjuvant. The same thing must also be explained for Ewing's sarcoma.
Response 5: We would like to thank the reviewer for pointing out the parts of our results that were not clear. We have additionally described the details of the aforementioned issues in the methods section. It now reads “For the treatment strategy for our patients with localized osteosarcoma and Ewing’s sarcoma, surgical resection of the primary tumor with adequate margins was an essential component of the curative strategy for patients with localized disease. Therefore, the chemotherapy for both osteosarcoma and Ewing’s sarcoma might start before (neoadjuvant) or after (adjuvant) curative surgical resection of the primary tumor. In our study, the patterns of chemotherapy prescription mainly depended on the schedule for a surgery, the adequacy of primary surgical margins, chemotherapy adverse effects, and the progression of disease following the initial chemotherapy treatment. Generally, almost all patients would receive neoadjuvant chemotherapy as their initial treatment or undergo curative surgery if it was available and not delayed by waiting to complete neoadjuvant chemotherapy treatments. If patients had completed neoadjuvant chemotherapy and definitive surgery with adequate surgical margins, the adjuvant chemotherapy usually would not be given after curative surgery. On the other hand, patients would be given adjuvant chemotherapy if they had not finished neoadjuvant chemotherapy before having curative surgery or if they had finished neoadjuvant chemotherapy but the surgical margins were not adequate. The adjuvant chemotherapy regimen would continue with the same regimen or a new regimen that included the same drugs used in the initial neoadjuvant chemotherapy unless there were adverse effects from the chemotherapy or the progression of disease during treatment.
For an unresectable tumor or metastasis, preoperative chemotherapy followed by surgery to remove the primary tumor and surgical resection of bone metastatic lesions followed by postoperative combination chemotherapy were employed as the primary approach. The alternative approach, starting with surgery for the primary tumor, followed by chemotherapy, and then surgical resection of metastatic bone lesions, would be employed if the primary approach was not appropriate (i.e., in patients with intractable pain, pathological fracture, or uncontrolled infection of the tumor, which could increase the risk of sepsis).
For radiation therapy, the objective of this treatment was local control. Radiation therapy was employed in the following cases: 1) localized resectable disease for which surgical margins could not be evaluated or at an inadequate surgical margin site; 2) unresectable disease; and 3) metastatic bone lesions.” (Page 4-5, Line 182-213)
Point 6: with such important biases from the chemotherapy point of view and with such a small sample size, I find it very difficult to highlight prognostic factors.
Response 6: We agree with the review’s point that a small sample size could make it difficult to highlight prognostic factors because of a low statistical power to detect an effect estimate and imprecision. However, we disagree that the reported effect estimates are biased from a small sample. It is possible that the effect estimates could be affected by residual/unadjusted confounders and time-varying variables. To minimize this bias on effect estimate, we thereby included the potential prognostic factors based on available evidence, including age, sex, received radiation therapy, disease status, and post neoadjuvant chemotherapy response; tumor necrosis rate (for osteosarcoma), and performed the survival analysis using the flexible parametric model that also adjusted for the time-varying effect of post neoadjuvant chemotherapy response (for osteosarcoma). Because the aim of our study is to quantify the effect of potential prognostic factors as unbiasedly and precisely as possible from the available observational data, we believe that the solution to observational analyses with imprecise effect estimates is not to avoid observational analyses with imprecise estimates, but rather to encourage the report of many observational analyses. As more studies become available, a meta-analysis can be conducted to provide a more precise estimate of the pooled effect.
In addition, we have calculated the minimum detectable hazard ratio based on the available sample size, observed event probability, and statistical power of 80%, using the power cox function via STATA16 to assure that the reported hazard ratio of significant prognosis factors is able to detect in our study as shown in the table below.
|
Patient groups |
Event probability (%) |
Minimum detectable hazard ratio |
|
|
Lower |
Upper |
||
|
Osteosarcoma (n=79) |
|
|
|
|
5-year Disease free survival |
39.6 |
≤ 0.37 |
≥ 2.72 |
|
5-year Overall survival |
45.5 |
≤ 0.39 |
≥ 2.54 |
|
Ewing’s sarcoma (n=23) |
|
|
|
|
5-year Disease free survival |
40.9 |
≤ 0.16 |
≥ 6.21 |
|
5-year Overall survival |
45.5 |
≤ 0.17 |
≥ 5.65 |
References:
Hernán M. A. (2022). Causal analyses of existing databases: no power calculations required. Journal of clinical epidemiology, 144, 203–205. https://doi.org/10.1016/j.jclinepi.2021.08.028
Reviewer 2 Report
Review for Cancers Paper
- Abstract: there are a lot of variables mentioned in the “study aimed”
first sentence, so reduce to overall key foci of study. Use evidence-
based, informed decision to decide what key variables investigating,
rather than adding multiple variables just because have data.
- Very well-written introduction
- Overall, excellent writing style and like the work- just merits some
adjusting of approach and analyses (see next sections)
- Apply same suggestion to rest of paper- narrow focus to key primary
variables, and add non-central variables as secondary variables
- Next, use the primary and secondary variables to guide paper order
and presentation (not other way around where just look at sampling of
data just because you have the data).
- Key feedback: when dividing a dataset by this many categories for
analysis, really reduce power, so decide categories and analysis
carefully, based on theory and evidence.
- For example, 23 people with Ewing’s is not really adequate sample size
for most analyses
o Then if you divide Ewing sample by sex, have negligible data
amount, not to mention you then analyze based on treatment,
palliative, etc.
o Might need to remove Ewing’s data, or include as secondary
analysis/focus
- How do you operationally define “at presentation” for paper (e.g., line
143)? This is critical for determining timeline, including replication of
study or comparison with other studies/datasets
- Line 135: What does “were eligible for the ascertainment of the study
outcomes” mean? Clarify as this is important for analysis/non-biased
approach.
- Quite a reduction in sample size and 23 Ewing’s = cannot compare
Ewing’s and osteo, nor do you have enough people in Ewing’s group
to subdivide by the different variables you mention as categories?
- Line 144: risk of disease occurrence should be further
described/defined.
o 2003 paper = is this standard paper according to which all
data for Ewing’s and osteo bases the disease risk? Such impt
for characterizing data so discuss more and describe
implications
- Line 180: the analysis should be conservative since have limited
sample size when divide by so many factors. Might be okay, just
address the potential implications of relatively small sample size when
divided by so many factors.
- How does your sample/data compare to larger, standardized datasets
like SEER, US, etc.?
- Figures starting at line 259 are too small and different font; that said,
I love the way the information is presented
- Table 4 is confusing, especially relative to the text that goes with it.
- I see pain, fatigue, fever in lines 266-69, but do not see them in
table 4
- If going to focus some emphasis on Phase, discuss more/introduce
why more in text beforehand
- Line 265 = how do you get n =181? I do not see a total of 181 in
Table 4
- Lines 266-69, make clear if n = patients or chemos
- Make sure tables are clear, and explained clearly in text (without
simply duplicating information)
- What does “responder” mean? Any analysis or results variables/factors
should be explained prior to analysis section
- Discussion and conclusions: try not to overstate your findings or
describe in too much detail, as these are small groups when subdivide
based on patient characteristics and/or treatment variables
- References: many of the references are 10+ years old; this can be
okay since there is not much data per your study aim, but seek newer
data to inform your study approach, using comparable studies.
Author Response
Point 1: Abstract: there are a lot of variables mentioned in the “study aimed” first sentence, so reduce to overall key foci of study. Use evidence-based, informed decision to decide what key variables investigating, rather than adding multiple variables just because have data.
Response 1: We would like to thank the reviewer’s suggestion. We have removed the outcome “chemotherapy prescribing patterns and response” from the first sentence of the abstract. It now reads “This study aimed to assess survival outcomes, prognostic factors, and adverse events following chemotherapy treatment for osteosarcoma and Ewing’s sarcoma. This retrospective observational study was conducted to collect the data of the patients with osteosarcoma or Ewing’s sarcoma who received chemotherapy treatment between 2008 and 2019. The flexible parametric survival model was performed to explore the adjusted survival probability and the prognostic factors. A total of 102 patients (79 with osteosarcoma and 23 with Ewing’s sarcoma) were included. The estimated 5-year disease-free survival (DFS) and 5-year overall survival (OS) probabilities in patients with a resectable disease were 60.9% and 63.3% for osteosarcoma, and 54.4% and 88.3% for Ewing’s sarcoma, respectively, whereas the 5-year DFS and 5-year OS for those with an unresectable/metastatic disease remained below 25%. Two prognostic factors for osteosarcoma included response to neoadjuvant chemotherapy and female gender. Ewing's sarcoma patients aged 25 years and older were significantly associated with poorer survival outcomes. Of 181 chemotherapy treatment cycles, common self-reported adverse symptoms included tumor pain (n = 32, 17.7%), fever (n = 21, 11.6%), and fatigue (n = 16, 8.8%), while common grade III adverse events included febrile neutropenia (n = 13, 7.3%) and neutropenia (n = 9, 5.1%). There was no chemotherapy-related mortality (grade V) or anaphylaxis events. (Page 1, Line 31-46).
Point 2: Very well-written introduction
Response 2: We would like to thank the reviewer for the suggestion and encouragement.
Point 3: Overall, excellent writing style and like the work- just merits some adjusting of approach and analyses (see next sections)
Response 3: We would like to thank the reviewer for the suggestion and encouragement.
Point 4: Apply same suggestion to rest of paper- narrow focus to key primary variables, and add non-central variables as secondary variables
Response 4: We would like to thank the reviewer for the valuable suggestion.
Point 5: Next, use the primary and secondary variables to guide paper order and presentation (not other way around where just look at sampling of data just because you have the data).
Response 5: As suggested by the reviewer, we have re-arranged the presentation of the method and the study result as follows:
- Materials and Methods
2.1. Study Design and Setting
2.2. Participants
2.3. Variables
2.4. Primary Outcomes: Survival Outcomes and Prosgnostic Factors
2.5. Secondary Outcomes: Prescribed Chemotherapy Regimens, Responses, and Adverse Effects
2.6. Treatment Strategy
2.7. Statistical Analysis
- Results (with subsections of osteosarcoma and Ewing’s sarcoma in each section)
3.1. Patient Characteristics
3.2. Survival Outcomes and Prognostic Factors
3.3. Chemotherapy Prescribing Patterns and Responses
3.4. Self-reported Symptoms and Adverse Events
Point 6: Key feedback: when dividing a dataset by this many categories for analysis, really reduce power, so decide categories and analysis carefully, based on theory and evidence.
Response 6: We would like to thank the reviewer for the valuable suggestion. We have recategorized the age groups of osteosarcoma and Ewing’s sarcoma as suggested in the reviewer’s point 11. We confirm that the categorization of age, response to chemotherapy, and types of disease for estimating the effect of the prognostic factors as the primary outcome are considered based on the ground theory and the established evidence. The rationale for these categories is described in section 2.3. Variables for age (Page 3, Line 145-147) and type of disease (Page 4, Line 148-154), and response to chemotherapy in section 2.5. Secondary outcomes: prescribed chemotherapy regimens, responses, and adverse effects (Page 4, Line 165-181).
Point 7: For example, 23 people with Ewing’s is not really adequate sample size for most analyses
Point 7.1: Then if you divide Ewing sample by sex, have negligible data amount, not to mention you then analyze based on treatment, palliative, etc.
Response 7 and 7.1: We recognize that 23 patients with Ewing’s sarcoma is a small sample size and might be inadequate for the particular analysis. However, we disagree that our analyses cannot be performed according to this limitation. For the analysis of the prognostic factors, the problem with the small sample size is the statistical power to detect “the effect” and imprecision (wide confident interval). We agree with the reviewer that more categories and variables in the model significantly decrease the statistical power, but it is just more difficult to determine a significant prognostic factor from the analyses. It does not mean that the reported effect estimates are biased and should not be reported. It is possible that the effect estimates could be affected by residual/unadjusted confounders and time-varying variables. To minimize this bias on effect estimate, we thereby included the potential prognostic factors based on the available evidence, including age, sex, received radiation therapy, disease status, and post neoadjuvant chemotherapy response; tumor necrosis rate (for osteosarcoma), and performed the survival analysis using the flexible parametric model that also adjusted for the time-varying effect of post neoadjuvant chemotherapy response (for osteosarcoma). Because the primary aim of our study is to quantify the effect of potential prognostic factors as unbiasedly and precisely as possible from the available observational data, we believe that the solution to observational analyses with imprecise effect estimates is not to avoid observational analyses with imprecise estimates, but rather to encourage the report of many observational analyses. As more studies become available, a meta-analysis can be conducted to provide a more precise estimate of the pooled effect.
In addition, we have calculated the minimum detectable hazard ratio based on the available sample size, observed event probability, and statistical power of 80%, using the power cox function via STATA16 to assure that the reported hazard ratio of significant prognosis factors is able to detect in our study as shown in the table below.
|
Patient groups |
Event probability (%) |
Minimum detectable hazard ratio |
|
|
Lower |
Upper |
||
|
Osteosarcoma (n=79) |
|
|
|
|
5-year Disease free survival |
39.6 |
≤ 0.37 |
≥ 2.72 |
|
5-year Overall survival |
45.5 |
≤ 0.39 |
≥ 2.54 |
|
Ewing’s sarcoma (n=23) |
|
|
|
|
5-year Disease free survival |
40.9 |
≤ 0.16 |
≥ 6.21 |
|
5-year Overall survival |
45.5 |
≤ 0.17 |
≥ 5.65 |
References:
- Hernán M. A. (2022). Causal analyses of existing databases: no power calculations required. Journal of clinical epidemiology, 144, 203–205. https://doi.org/10.1016/j.jclinepi.2021.08.028
Point 7.2: Might need to remove Ewing’s data, or include as secondary analysis/focus
Response 7.2: Because of a constraint on a table and a figure limit in the journal, as well as the quantity of information, it is unable to present the results of osteosarcoma and Ewing’s sarcoma separately. To make the presentation clearer, we revised the main texts in the result sections by separately describing the results of osteosarcoma and Ewing's sarcoma in separate subsections, except for self-reported symptoms and adverse events.
Point 8: How do you operationally define “at presentation” for paper (e.g., line 143)? This is critical for determining timeline, including replication of study or comparison with other studies/datasets
Response 8: We would like to apologize for our mistake. We have corrected the sentence. It now reads “. Patients were divided into two groups based on their staging status at diagnosis (localized or metastatic disease).” (Page 3-4, Line 147-148)
Point 9: Line 135: What does “were eligible for the ascertainment of the study outcomes” mean? Clarify as this is important for analysis/non-biased approach.
Response 9: We would like to apologize for this unclear statement. The “eligible for the ascertainment of the study outcomes” means The patient’s electronic medical record data meets inclusion criteria for review and analysis of the study outcomes. We have rephrased the sentence to clarify as this is important for an analysis/non-biased approach. It now reads “Of a total of 324 cases diagnosed with bone and soft tissue sarcoma, 79 cases of osteosarcoma and 23 cases of Ewing’s sarcoma met the inclusion criteria for review and analysis of the study outcomes.” (Page 3, Line 136-138).
Point 10: Quite a reduction in sample size and 23 Ewing’s = cannot compare Ewing’s and osteo, nor do you have enough people in Ewing’s group to subdivide by the different variables you mention as categories?
Response 10: We have clarified these issues in the response to point 7 and 7.1. for the primary outcomes on survival outcome and prognostic factors.
For the categories in the descriptive results, the findings on patients’ characteristics, treatment patterns, and chemotherapy responses are presented to provide the details of our study context for the evaluation of its external validity by readers. Because the reported data could be different or similar to other studies by chance due to a small sample size, the comparison of our patients’ characteristics, prescribed chemotherapy patterns, and chemotherapy responses to other studies, particularly for Ewing’s sarcoma, should be done with caution and awareness of this limitation. We have added this limitation to the discussion. It now reads “Finally, because the reported descriptive data could be under detected due to the small sample size, the comparison of our patients’ characteristics, prescribed chemotherapy patterns, and chemotherapy responses to other studies, particularly for Ewing’s sarcoma, should be done with caution and awareness of this limitation.” (Page 15, Line 467-470)
Point 11: Line 144: risk of disease occurrence should be further described/defined.
o 2003 paper = is this standard paper according to which all data for Ewing’s and osteo bases the disease risk? Such impt for characterizing data so discuss more and describe implications
Response 11: We apologize for our mistakes in the presented information. According to the reviewer’s suggestion, we have looked for recent comparable evidence for the patient’s age and clinical outcomes in both osteosarcoma and Ewing’s sarcoma. We agree with the review that the age group in our study should be recategorized based on the established evidence. We have recategorized the age group as <18 and ≥18 years for osteosarcoma [ref.1] and <25 and ≥ 25 years for Ewing’s sarcoma [ref.2]. For osteosarcoma, patients in the older adolescent and young adult age groups ( ≥ 18 years) tend to have a worse prognosis due to a higher rate of relapse [ref.1]. For Ewing’s sarcoma, younger patients have a better prognosis than adolescents and adult patients aged 25 years and older in terms of metastatic disease at diagnosis and unfavorable sites of the tumor, and higher tumor volume [ref.2].
We have revised the description of age categories. It now reads “We categorized the age group as < 18 and ≥ 18 years for osteosarcoma and < 25 and ≥ 25 years for Ewing’s sarcoma. The age categories were decided based on the evidence as reported elsewhere [39][40]. ” (Page 3, Line 145-147)
Then, we reanalyzed the prognostic factors using the new age groups. As a result, the reanalyzed results of the prognostic factors remained the same as our previous result, with slightly changed effect estimates except for the new age group of Ewing’s sarcoma. Ewing’s sarcoma patients, age ≥ 25 years, showed significantly poorer both 5-year DFS and 5-year OS. We have added further discussion on the association between age groups and survival outcomes in osteosarcoma (Page 14, Line 421-426) and Ewing’s sarcoma (Page 14, Line 426-433).
References:
- Janeway, K. A., Barkauskas, D. A., Krailo, M. D., Meyers, P. A., Schwartz, C. L., Ebb, D. H., Seibel, N. L., Grier, H. E., Gorlick, R., & Marina, N. (2012). Outcome for adolescent and young adult patients with osteosarcoma: a report from the Children's Oncology Group. Cancer, 118(18), 4597–4605. https://doi.org/10.1002/cncr.27414
- Worch, J., Ranft, A., DuBois, S. G., Paulussen, M., Juergens, H., & Dirksen, U. (2018). Age dependency of primary tumor sites and metastases in patients with Ewing sarcoma. Pediatric blood & cancer, 65(9), e27251. https://doi.org/10.1002/pbc.27251
Point 12: Line 180: the analysis should be conservative since have limited sample size when divide by so many factors. Might be okay, just address the potential implications of relatively small sample size when divided by so many factors.
Response 12: We have clarified these issues about the analysis of the survival outcome and prognostic factors. Please see the response to point 7 and 7.1. We have addressed the rationale of our analyses in section 2.7. Statistical analysis as suggested by the reviewer. It now reads “To quantify the effect of potential prognostic factors as unbiasedly and precisely as possible, the potential prognostic factors were chosen based on the ground theory and the established evidence, including age, sex, radiation therapy, disease status, and post-neoadjuvant chemotherapy response (for osteosarcoma). The survival model for osteosarcoma was also adjusted for the time-varying effect of post-neoadjuvant chemotherapy response.” (Page 5, Line 223-227).
Point 13: How does your sample/data compare to larger, standardized datasets like SEER, US, etc.?
Response 13: Our study size is very small when compared with the reported data on large cancer database, such as SEER (5016 osteosarcoma patients between 1975-2017 [ref.1] and 3162 Ewing’s sarcoma patients between 2004 and 2015 [ref.2]). However, there is not much difference when compared to a sample size of individual studies from large osteosarcoma research groups in America and Europe, ranging from 51 to 677, but most studies are less than 200 as reported in the meta-analysis [ref.3]. For the studies from experience centers, large Ewing’s sarcoma research groups (i.e., Memorial Sloan-Kettering Cancer Center, COG, EURO-EWING), a sample size ranges from 64 to 518 as summarized in this review article [ref.4].
References:
- Cole, S., Gianferante, D. M., Zhu, B., & Mirabello, L. (2022). Osteosarcoma: A Surveillance, Epidemiology, and End Results program-based analysis from 1975 to 2017. Cancer, 128(11), 2107–2118. https://doi.org/10.1002/cncr.34163
- Brown, J. M., Rakoczy, K., Tokson, J. H., Jones, K. B., & Groundland, J. S. (2022). Ewing sarcoma of the pelvis: Clinical features and overall survival. Cancer treatment and research communications, 33, 100634. https://doi.org/10.1016/j.ctarc.2022.100634
- Anninga, J. K., Gelderblom, H., Fiocco, M., Kroep, J. R., Taminiau, A. H., Hogendoorn, P. C., & Egeler, R. M. (2011). Chemotherapeutic adjuvant treatment for osteosarcoma: where do we stand?. European journal of cancer (Oxford, England : 1990), 47(16), 2431–2445. https://doi.org/10.1016/j.ejca.2011.05.030
- Zöllner, S. K., Amatruda, J. F., Bauer, S., Collaud, S., de Álava, E., DuBois, S. G., Hardes, J., Hartmann, W., Kovar, H., Metzler, M., Shulman, D. S., Streitbürger, A., Timmermann, B., Toretsky, J. A., Uhlenbruch, Y., Vieth, V., Grünewald, T. G. P., & Dirksen, U. (2021). Ewing Sarcoma-Diagnosis, Treatment, Clinical Challenges and Future Perspectives. Journal of clinical medicine, 10(8), 1685. https://doi.org/10.3390/jcm10081685
Point 14: Figures starting at line 259 are too small and different font; that said, I love the way the information is presented
Response 14: We would like to thank the reviewer for the suggestion and the positive feedback. We have increased the font size and changed the font style in the revised figures 1 (Page 7-8, Line 280-281) and figure 2 (Page 9-10, Line 295-296).
Point 15: Table 4 is confusing, especially relative to the text that goes with it. I see pain, fatigue, fever in lines 266-69, but do not see them in table 4
Response 15: To reduce the amount of information in Table 4 that could make readers confused, we have separated the results of adverse drug reactions following chemotherapy treatment into Table 5 (Page 12-13, Line 347-348). We also checked the missing information as mentioned by the reviewer and the data were highlighted in the column “Self-reported adverse symptoms” in table 4 (Page 12, Line 345-346).
Point 16: If going to focus some emphasis on Phase, discuss more/introduce why more in text beforehand
Response 16: We have added the explanation of why we focused on self-reported symptoms and adverse drug reactions following chemotherapy treatment by the number of specific chemotherapy drugs in each cycle.
“Due to the fact that there were a variety of prescribed regimens in our study and that each cycle typically included a specific combination of chemotherapy that can cause different or overlapping adverse effects, it was difficult to determine which single agent was the causative agent, especially for common symptoms and adverse events. In addition, an increase in the number of specific chemotherapy drugs in each cycle may improve clinical outcomes, while increasing the risk of adverse effects. Thus, the self-reported adverse symptoms and adverse events are summarized by the number of specific chemotherapy drugs in each cycle in Table 4 and Table 5, respectively.” (Page 12, Line 327-334)
Point 17: Line 265 = how do you get n =181? I do not see a total of 181 in Table 4
Response 17: We would like to apologize for our unclear statement. Recorded data showed that 181 chemotherapy cycles resulted in self-reported symptoms by the patients who received them. We have added this sentence to clarify this number. It now reads “Recorded data showed that 181 chemotherapy cycles resulted in self-reported symptoms by the patients who received them.” (Page 12, Line 334-339).
Point 18: Lines 266-69, make clear if n = patients or chemos
Response 18: We would like to apologize for our unclear statement. “n” represents the number of chemotherapy cycles. To make it clearer, we have added more details. It now reads “Recorded data showed that 181 chemotherapy cycles resulted in self-reported symptoms by the patients who received them. Of the 181 cycles provided, common self-reported adverse symptoms included tumor pain (32 cycles, 17.7%), fever (21 cycles, 11.6%), and fatigue (16 cycles, 8.8%). Common grade III adverse events included febrile neutropenia (13 cycles, 7.3%) and neutropenia (9 cycles, 5.1%).” (Page 12, Line 334-339).
Point 19: Make sure tables are clear, and explained clearly in text (without simply duplicating information)
Response 19: We have thoroughly reviewed our results once again. We have revised the results’ presentation and description based on the reviewers’ suggestions. We hope that the revised manuscript will meet the reviewers' satisfaction.
Point 20: What does “responder” mean? Any analysis or results variables/factors should be explained prior to analysis section
Response 20: We agree that any analysis or results variables/factors should be explained prior to the analysis section. The term “responder” was defined in the methods section as follows: “1) Neoadjuvant chemotherapy: chemotherapy given to patients before surgical resection of the primary tumor. The response to neoadjuvant chemotherapy was determined by a tumor necrosis rate (TNR) from the post-operative histopathological report. According to the Huvos grading system[20], those with a TNR between 90% and 99% (grade III) or 100% (grade IV) were considered responders, whereas those with a TNR of less than 50% (grade I) or between 50% and 89% (grade II) were considered non-responders.” (Page 4, Line 165-171).
Point 21: Discussion and conclusions: try not to overstate your findings or describe in too much detail, as these are small groups when subdivide based on patient characteristics and/or treatment variables
Response 21: We would like to thank the reviewer for the suggestion. We have thoroughly reviewed our discussion and conclusion once again. We assure the reviewer that the discussion and conclusion on the finding are not overstated as we clarified in response to the reviewer on point 7 and point 10.
Point 22: References: many of the references are 10+ years old; this can be okay since there is not much data per your study aim, but seek newer data to inform your study approach, using comparable studies.
Response 22: As the reviewer suggested, we looked for articles and more recent data. We have cited recent studies on the prognostic factors (age groups and sex) in osteosarcoma and Ewing’s sarcoma, but no further comparable studies in a similar setting as ours were found.
As we stated in the introduction, the main reason why we conducted this study is that most of the available evidence is only from large cancer centers in developed countries. These data are already established and not novel at all for their contexts, but not for settings like ours, where the information on the longitudinal outcomes following chemotherapy in patients with bone sarcoma is still limited.
Reviewer 3 Report
The goal of the manuscript is to assess the adjusted survival probability and the prognostic factors of Osteosarcoma (79 patients) and Ewing sarcoma (23 patients). The Overall Survival (OS), Disease Free Survival (DFS), were assessed by comparing the patients who received nonadjuvant chemotherapy or no-nonadjuvant chemotherapy, who were respectable or unresectable status, who received radiation or no-radiation. In addition, the patients of OS and DFS were modeled by their age and gender. The manuscript also includes the adverse effect of both diseases. While the manuscript supplies an important information, it provides a modest level of information to the field because most of the conclusions are similar to the previous reports. The specific comments for the manuscript are listed below.
- Abstract should include the conclusion of the report.
- Describe the race of the patients. If the patients are predominantly Asian population, the study would add a new information to the field because the outcomes of the has been predominantly studied in Caucasian population.
- OS and DFS of female patients with Osteosarcoma are significantly higher than male patients, whereas Ewing sarcoma patients does not show the significance in its change. Is it possible that the low numbers of Ewing sarcoma patients contributing to the result? Or is this indicating the difference between Osteosarcoma or Ewing sarcoma? Discuss the difference between two diseases.
Author Response
Point 1: The goal of the manuscript is to assess the adjusted survival probability and the prognostic factors of Osteosarcoma (79 patients) and Ewing sarcoma (23 patients). The Overall Survival (OS), Disease Free Survival (DFS), were assessed by comparing the patients who received neoadjuvant chemotherapy or no-neoadjuvant chemotherapy, who were respectable or unresectable status, who received radiation or no-radiation. In addition, the patients of OS and DFS were modeled by their age and gender. The manuscript also includes the adverse effect of both diseases. While the manuscript supplies an important information, it provides a modest level of information to the field because most of the conclusions are similar to the previous reports. The specific comments for the manuscript are listed below.
Response 1: We would like to thank the reviewer for the suggestion and encouragement.
Point 2: Abstract should include the conclusion of the report.
Response 2: We have added a sentence to the abstract. It now reads “This study aimed to assess survival outcomes, prognostic factors, and adverse events following chemotherapy treatment for osteosarcoma and Ewing’s sarcoma. This retrospective observational study was conducted to collect the data of the patients with osteosarcoma or Ewing’s sarcoma who received chemotherapy treatment between 2008 and 2019. The flexible parametric survival model was performed to explore the adjusted survival probability and the prognostic factors. A total of 102 patients (79 with osteosarcoma and 23 with Ewing’s sarcoma) were included. The estimated 5-year disease-free survival (DFS) and 5-year overall survival (OS) probabilities in patients with a resectable disease were 60.9% and 63.3% for osteosarcoma, and 54.4% and 88.3% for Ewing’s sarcoma, respectively, whereas the 5-year DFS and 5-year OS for those with an unresectable/metastatic disease remained below 25%. Two prognostic factors for osteosarcoma included response to neoadjuvant chemotherapy and female gender. Ewing's sarcoma patients aged 25 years and older were significantly associated with poorer survival outcomes. Of 181 chemotherapy treatment cycles, common self-reported adverse symptoms included tumor pain (n = 32, 17.7%), fever (n = 21, 11.6%), and fatigue (n = 16, 8.8%), while common grade III adverse events included febrile neutropenia (n = 13, 7.3%) and neutropenia (n = 9, 5.1%). There was no chemotherapy-related mortality (grade V) or anaphylaxis events.”(Page 1, Line 31-46).
Point 3: Describe the race of the patients. If the patients are predominantly Asian population, the study would add a new information to the field because the outcomes of the has been predominantly studied in Caucasian population.
Response 3: All of the included patients were Asian. We have added this detail according to the reviewer’s suggestion. It now reads “All of the participants were Asian.” (Page 5, Line 234).
Point 4: OS and DFS of female patients with Osteosarcoma are significantly higher than male patients, whereas Ewing sarcoma patients does not show the significance in its change. Is it possible that the low numbers of Ewing sarcoma patients contributing to the result? Or is this indicating the difference between Osteosarcoma or Ewing sarcoma? Discuss the difference between two diseases.
Response 4: We would like to thank the reviewer for pointing this out. From the previous literature, the female gender has been associated with a better prognosis for both osteosarcoma and Ewing’s sarcoma. Because sex has a relatively minor effect on the survival outcome when compared to other factors, the discordance in our finding on the prognostic factors for Ewing’s sarcoma, especially sex, may be difficult to detect due to the small number of Ewing’s sarcoma cases.
Because osteosarcoma and Ewing's sarcoma have two completely different pathways from the point of view of pathology and prognosis, a comparison between the two diseases may not be appropriate.
References:
- Smeland, S., Bielack, S. S., Whelan, J., Bernstein, M., Hogendoorn, P., Krailo, M. D., Gorlick, R., Janeway, K. A., Ingleby, F. C., Anninga, J., Antal, I., Arndt, C., Brown, K. L. B., Butterfass-Bahloul, T., Calaminus, G., Capra, M., Dhooge, C., Eriksson, M., Flanagan, A. M., Friedel, G., … Marina, N. (2019). Survival and prognosis with osteosarcoma: outcomes in more than 2000 patients in the EURAMOS-1 (European and American Osteosarcoma Study) cohort. European journal of cancer (Oxford, England : 1990), 109, 36–50. https://doi.org/10.1016/j.ejca.2018.11.027
- Janeway, K. A., Barkauskas, D. A., Krailo, M. D., Meyers, P. A., Schwartz, C. L., Ebb, D. H., Seibel, N. L., Grier, H. E., Gorlick, R., & Marina, N. (2012). Outcome for adolescent and young adult patients with osteosarcoma: a report from the Children's Oncology Group. Cancer, 118(18), 4597–4605. https://doi.org/10.1002/cncr.27414
- Collins, M., Wilhelm, M., Conyers, R., Herschtal, A., Whelan, J., Bielack, S., Kager, L., Kühne, T., Sydes, M., Gelderblom, H., Ferrari, S., Picci, P., Smeland, S., Eriksson, M., Petrilli, A. S., Bleyer, A., & Thomas, D. M. (2013). Benefits and adverse events in younger versus older patients receiving neoadjuvant chemotherapy for osteosarcoma: findings from a meta-analysis. Journal of clinical oncology : official journal of the American Society of Clinical Oncology, 31(18), 2303–2312. https://doi.org/10.1200/JCO.2012.43.8598
- Jawad, M. U., Cheung, M. C., Min, E. S., Schneiderbauer, M. M., Koniaris, L. G., & Scully, S. P. (2009). Ewing sarcoma demonstrates racial disparities in incidence-related and sex-related differences in outcome: an analysis of 1631 cases from the SEER database, 1973-2005. Cancer, 115(15), 3526–3536. https://doi.org/10.1002/cncr.24388
- Bacci, G., Longhi, A., Ferrari, S., Mercuri, M., Versari, M., & Bertoni, F. (2006). Prognostic factors in non-metastatic Ewing's sarcoma tumor of bone: an analysis of 579 patients treated at a single institution with adjuvant or neoadjuvant chemotherapy between 1972 and 1998. Acta oncologica (Stockholm, Sweden), 45(4), 469–475. https://doi.org/10.1080/02841860500519760
- Karski, E. E., McIlvaine, E., Segal, M. R., Krailo, M., Grier, H. E., Granowetter, L., Womer, R. B., Meyers, P. A., Felgenhauer, J., Marina, N., & DuBois, S. G. (2016). Identification of Discrete Prognostic Groups in Ewing Sarcoma. Pediatric blood & cancer, 63(1), 47–53. https://doi.org/10.1002/pbc.25709
Round 2
Reviewer 1 Report
the corrections made by the authors are relevant to the suggestions